# STL: Still Tricky Logic (for System Validation, Even When Showing Your Work)

**Isabelle Hurley**[*]
Lincoln Laboratory
Massachusetts Institute of Technology
Lexington, MA 02421
isabelle.hurley@ll.mit.edu

**Rohan Paleja**
Lincoln Laboratory
Massachusetts Institute of Technology
Lexington, MA 02421
rohan.paleja@ll.mit.edu

**Ashley Suh**
Lincoln Laboratory
Massachusetts Institute of Technology
Lexington, MA 02421
ashley.suh@ll.mit.edu

**Jaime D. Peña**
Lincoln Laboratory
Massachusetts Institute of Technology
Lexington, MA 02421
jdpena@ll.mit.edu

**Ho Chit Siu**
Lincoln Laboratory
Massachusetts Institute of Technology
Lexington, MA 02421
hochit.siu@ll.mit.edu

## Abstract

As learned control policies become increasingly common in autonomous systems, there is increasing need to ensure that they are interpretable and can be checked by human stakeholders. Formal specifications have been proposed as ways to produce human-interpretable policies for autonomous systems that can still be learned from examples. Previous work showed that despite claims of interpretability, humans are unable to use formal specifications presented in a variety of ways to validate even simple robot behaviors. This work uses active learning, a standard pedagogical method, to attempt to improve humans' ability to validate policies in signal temporal logic (STL). Results show that overall validation accuracy is not high, at $65\% \pm 15\%$ (mean $\pm$ standard deviation), and that the three conditions of no active learning, active learning, and active learning with feedback do not significantly differ from each other. Our results suggest that the utility of formal specifications for human interpretability is still unsupported but point to other avenues of development which may enable improvements in system validation. [2]

## 1 Introduction

In recent years, research in robotics has made incredible strides and has been targeting applications across a variety of industries, including manufacturing, healthcare, household, and security. Across

---

[*]Corresponding author.

[2]DISTRIBUTION STATEMENT A. Approved for public release. Distribution is unlimited.

This material is based upon work supported by the Under Secretary of Defense for Research and Engineering under Air Force Contract No. FA8702-15-D-0001. Any opinions, findings, conclusions or recommendations expressed in this material are those of the author(s) and do not necessarily reflect the views of the Under Secretary of Defense for Research and Engineering.

38th Conference on Neural Information Processing Systems (NeurIPS 2024).

these diverse use cases, a central envisioned capability is that of dynamic programming of robots, allowing an end-user — typically with less familiarity with the system than the original designer — to modify the robotic system to suit their own purposes, such as in home robotics applications [37, 40], agricultural monitoring [5], or Internet of Things devices [12, 21]. However, prior to such a capability, robots must be able to perform tasks safely, and the programmer must be able to effectively ensure that the robot's behavior and its respective changes meet the intended goal.

In quality control, this process is commonly referred to as Verification & Validation (V&V). Specifically, properties and behaviors of robotic systems must meet a set of broader requirements (i.e., human safety, legal standards, etc.) and align with the intent of stakeholders. Depending on the techniques used to create robot behavior, the former (verification) can be accomplished via sampling and simulation, or even via mathematical proofs [13, 11]. The latter (validation) is a process that determines whether the product "solves the right problems" and "satisf[ies] intended use and user needs" [11], making it an *inherently human-centered process of value alignment*. Validation is a key building block in creating robotic systems that are able to be dynamically reprogrammed and building a common language between the user and robot.[3]

While significant work has been performed on system verification [19, 20], much less has been done on validation. Much of contemporary verification work relies on the use of formal methods, including languages such as linear temporal logic (LTL) and signal temporal logic (STL). There is also an assumption in the formal methods community that these methods are not only verifiable, but are also human-interpretable, and thus useful for validation due to their semantics (e.g. [6, 26, 14, 28, 10, 23]). However, human validation of policies expressed in both LTL and STL has shown to be empirically very difficult [17, 38], even as it is essential. Indeed, as Leveson observes, "virtually all accidents involving software stem from unsafe requirements, not implementation errors," and "software logic flaws stem from a lack of adequate requirements" [24].

Given the importance of validation and these difficulties in interpreting autonomous system behavior specified via formal logic, we propose the use of *active learning*, a popular educational practice, to improve humans' ability to validate formal logic policies. This active learning is distinct from active learning in the machine learning literature, and is a human-focused activity (detailed in Section 2.2). Our primary research question is whether or not using active learning improves humans' policy validation rate. In this paper, we contribute 1) a (human) active learning method as a potential approach to improving humans' ability to interpret formal policies, leveraging tools common in formal methods, 2) an experimental protocol to measure the differences in system validation performance under varying active learning conditions, and 3) a set of implications for improving human validation of autonomous systems. In short, formal specifications are not inherently human interpretable for system validation and while active learning approaches can improve human engagement with a system validation task, this does not necessarily improve performance.

## 2 Background

### 2.1 Formal Methods for Interpretable Learning

Autonomous systems can be programmed in a variety of ways, including codifying domain expertise, extracting policies from demonstration [9, 35], or synthesizing behavior via an objective function and constraints [25]. Policies can be represented in a variety of forms, including neural networks, symbolic representations, and behavior trees. In this paper, we focus on a popular representation, temporal logic, a subset of formal methods. Temporal logic has seen significant usage with autonomous systems (e.g. [6, 26, 14, 28, 10, 23]) as it can represent high-level mission objectives, allowing for ease of programming to those familiar with its use. Furthermore, these mathematically precise specifications allow for representing model behavior within a relatively concise format and verification to ensure the autonomous system behavior meets requirements. A more complete introduction to STL and its use in this experiment is in Appendix A.1.

---

[3]Some definitions of validation, such as from system modeling [13] are slightly different, and in some cases there is disagreement about whether verification and validation are distinct activities. We take the software engineering standard definition from the IEEE [11], which distinguishes the two and is more appropriate for autonomous systems design.

Human stakeholders' ability to understand policies (to varying degrees) is increasingly important from legal, ethical, and usability perspectives, particularly if policies can be updated after system deployment, either in a centralized manner, or in a user-initiated manner. *Interpretability*, and related terms such as *transparency*, *explainability*, and so on, are, however, often poorly defined, or indeed, simply left undefined in the research that claim to have created systems with these properties [27]. Miller et al.'s 2017 work showed that among a collection of papers from an Explainable AI workshop, most papers neither cited supporting social science literature to support their design choices, nor reached their conclusions with actual data showing any measure of interpretability; indeed "many models being used in explainable AI research are not building on current scientific understanding of explanation" [32]. Sanneman and Shah propose the use of *situation awareness* to determine the informational needs driving requirements of explainable systems [36]. User informational needs map to the levels of *perception*, *comprehension*, or *prediction* of system behavior. For our purposes of system validation, a strong ability to *predict* behavior is the kind of interpretability required to ensure intent alignment.

Formal methods, which employ logical languages, are a collection of system design techniques that rigorously tie semantic properties to mathematical models, and have been touted as a promising approach to interpretable policies, in addition to their mathematical verifiability [6, 26, 14, 28, 10, 23]. The supposed link to interpretability (almost universally left unstated) is likely that if model behaviors can be tied specifically to meaningful, grounded concepts (such as symbols and prepositions), *and* if models are small enough for humans to reasonably examine, then interpretability is achieved. Indeed, unlike "black box" methods such as deep neural networks, the semantics of formal logic ensures that their specification can always be expressed in natural language.

The major gap in these approaches to interpretability is the lack of empirical evidence to their efficacy in the overwhelming majority of cases. For example [6, 26, 14, 28, 10, 23] all claim to have methods that learn human-interpretable policies via formal methods. All provide empirical evidence that policies are learned, *but all fail to provide any empirical evidence for the human-interpretability of their policies*. A more extensive 2023 survey of the last decade of temporal logic literature that specifically claimed interpretability saw that only approximately 10% cited any supporting work for the interpretability of their methods, none actually incorporated the cited methods into their work, and none checked their claims with actual humans [38]. Subsequent human experiments examining interpretability for validation in that work showed less than 50% validation accuracy with signal temporal logic, even when using methods proposed by the literature (e.g. translation into language or decision trees). Formal methods experts only performed marginally better than complete novices. Those results match what is found in a broader 2023 review of the existing empirical work on explainable AI, where human performance has been extremely poor, even when self-reported understanding and trust increases [18].

While methods like translation into language may seem an intuitive way to increase the interpretability of formal logic, empirical evidence does not demonstrate this improvement [38, 39]. Vinter et al. [39] and Loomes and Vinter [29] showed that specifications rendered in natural language (even when containing logical operators) evoked inappropriate systemic biases in which readers substituted heuristic reasoning (commonly used in language) for logical reasoning (necessary for formal methods) during evaluation. Consequently, we do not provide a translation into natural language for this experiment.

In this paper, we aim to harness human active learning to bridge the gap between claims of interpretability and the actual understanding of users. Our goal is to provide users with a deeper insight into the behavior of autonomous agents as specified by Signal Temporal Logic (STL). A detailed description of the STL format utilized for all specifications can be found in Appendix A.2.

## 2.2 Human Active Learning for Validation

Following Miller et al.'s exhortation [32], we explore the utility of current scientific understanding of human learning in the process of interpreting robot policies for validation. *Active learning* as scoped by Bonwell and Eison, involves "instructional activities involving students in doing things and thinking about what they are doing," and involves "higher-order thinking tasks as analysis, synthesis, and evaluation" [2]. Common examples of active learning activities in the classroom include note-taking, note synthesis, writing exercises, and discussion. Active learning has been shown to benefit learners by increasing their engagement in areas which require higher order thinking skills

like engineering [1] and a 2014 metaanalysis found that across 225 studies conducted within STEM courses, active learning increased learner performance and decreased rates of class failure [16].

In systems where users work with programmed robots, active learning can be applied as a tool to increase user understanding over robot behavior. To enable this higher learning for validation, we ask users to assess the contextual implications of a given policy and evaluate how the allowed set of behaviors aligns with one's needs. Under the adapted Bloom's Taxonomy of (human) learning [15], this assessment and evaluation would touch on higher levels of learning, specifically enabling the following dimensions of learning: Recognition, Assembly, Determination, and Judgement.

## 3 Methods

Providing humans with the ability to validate robot policies specified in STL is a building block toward dynamic reprogramming of robots. In this section, we describe how to incorporate active learning to improve a human's ability to perform system validation.

### 3.1 Experimental Testbed: ManeuverGame

To investigate how pedagogical active learning practices can help improve humans' ability to validate the policy of an autonomous system, we developed *ManeuverGame*, a software suite centered around policy evaluation for agents in a grid world game. *ManeuverGame* allows a user to control an agent and navigate it through a map given configuration. Trajectories can be easily generated, saved and iterated upon for users to explore the behavior constraints of a specification within the dynamics of a specific scenario. Thus, users can refine of their understanding of the policies by performing all aspects of Bloom's taxonomy [15] with given set of specifications. Figure 1 depicts this process.

#### 3.1.1 Active Learning For Validation Concept

Active learning and feedback can be integrated into the process of system validation. We approach this by tasking learners to generate example behaviors that satisfy the specification's constraints in order to support their determinations of a specification's validity. If a behavior can be found that meets the specification but nonetheless violates the user's intent, then it demonstrates that the specification is *invalid*. Figure 1 (left) shows an example of a specification used in this study.

Rather than solely tasking users with evaluating if a specification is valid or not, we investigate an active learning approach which breaks the validation of specifications into steps of first behavior exploration and secondly determination. In behavior exploration, users explore a subset of the allowed behaviors defined by the specification's constraints. In determination, users evaluate if the set of allowed behaviors contains only trajectories which match their intent or if it includes trajectories which violate their intent. Essentially, this approach asks users to explore the set of behaviors and provide specific examples to "show their work" in making a specification determination.

Runtime verification is a computational approach to detect if system behaviors match a specification [22]. Such techniques can be used to facilitate feedback about specific behaviors, since they are a lightweight alternative to other verification techniques like model checking and can be automatically generated from high level specifications [22]. A runtime verification *monitor* takes in a finite trace (i.e. a trajectory) and determines a correctness property. It can be applied in the context of active learning for validation to give users feedback on example behaviors (trajectories).

We implement our experiment in the custom *ManeuverGame* interface (Figure 1). An agent has a specification that determines the trajectories it can generate, and the subject must determine if the specification will cause the agent to always win regardless of what trajectory is generated. The blue agent (circle) wins if it reaches the green objective (triangle) within 30 steps, while avoiding being within one space of the static red players (diamond).

Where verification asks the question "does this product/behavior match with the set of requirements set out for it," validation probes "does this product operate in and only in the ways that I want it to?" In our experiment, "winning" serves as a clear objective stakeholder intent —— essentially defining "the ways I want it to" —— that can be codified in programming. This approach replaces the more complex and less easily codified intents often held by robotics stakeholders, enabling us to explore the limitations and challenges of human validation in a controlled environment.

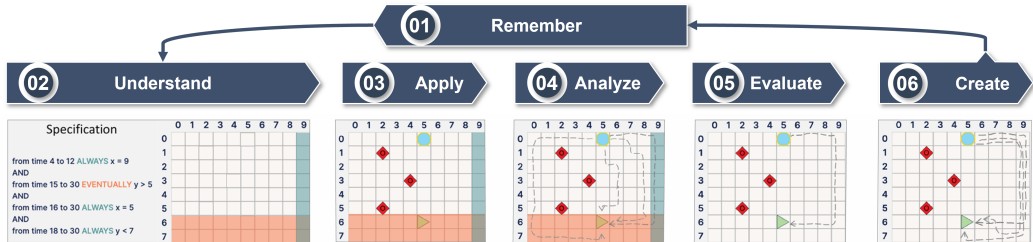

Figure 1: Bloom's taxonomy [15] applied to ManeuverGame. For a given problem, a subject must first recall the meaning of the information they're being presented with (e.g., a formal specification), and understand it in the given context (grid world). Applying these concepts in ManeuverGame enables the subject to analyze different trajectories, both valid and invalid, under a trial-and-error process. Evaluating the trajectories, a subject is able to hone in on specifications that are both valid and meet the specification, allowing them to create multiple such specifications. This process repeats itself for new problems.

Without presenting to the user a task that can be checked automatically, we cannot objectively check the user's ability to evaluate the specification. Thus, while assessing whether specifications lead to winning can be framed as a verification problem, it provides a clear representation of stakeholder intent, unlike the more complex intents that robotics stakeholders may have such as "complete the task safely" or "watch out for nearby cyclists." If such a simple intent cannot be easily validated, more difficult tasks are likely worse off.

## 4   Human-Subjects Study

We conducted a between-subjects experiment where human participants validated robot motion plans from provided trajectory specifications. 55 adult participants completed the experiment after providing informed consent. The protocol was approved by the MIT Committee on the Use of Humans as Experimental Subjects and the United States Department of Defense Human Research Protection Office. Participants were recruited from the MIT Behavioral Research Lab participant pool, which includes a diverse set of volunteers from the general public, along with a small number of targeted emails to recruit people with formal methods expertise. The pool was designed to reflect a likely range of "end users" of end-user-reprogrammable systems. Participants were compensated $15 USD for completion of the experiment plus $2 per correct validation, for a total of up to $35. Broadly, the experiment asked participants to determine whether a provided temporal logic specification would always result in plans that would win a capture-the-flag-like game in a provided game configuration. Below, we detail the experiment procedure.

### 4.1   Procedure

All subjects are first provided with a demographic survey focused on relevant educational background (e.g., experience in logic, mathematics, robotics), with 5-point Likert scale questions and open-ended explanations. Next, subjects received introductory material via videos, text-based tutorials and interactive questions as described in Appendix A.3.

Then, the participant shifted into the main experiment. During this phase, subjects were presented with a robot policy as a set of motion constraints (specifications) in STL alongside a starting map, and were asked to evaluate whether trajectories generated by the policy would *always* allow the blue agent to win the game (valid) or could result in losing (invalid), followed by their confidence in their answer. Each subject evaluated ten pairs of specifications and maps during the course of the experiment. For a sense of specification complexity across the evaluation tasks, the number of symbols in the specification varied from 43 to 97, and the abstract syntax tree (AST) depth ranged from 3 to 5.

During evaluation, both Active Learning groups (**AL-WF** and **AL-NF**) were required to either provide three trajectories that 1) satisfied the specification and resulted in a blue team win before answering that a specification was valid, or 2) provide one trajectory that satisfied the specification

but resulted in a blue team loss before answering that a specification was invalid. The former required users to explore potential implications of the specification, but does not represent an exhaustive search of the possibilities, while the latter is effectively a proof by contradiction. While subjects are asked to provide trajectories that meet the specifications, the **AL-NF** group was not provided real-time feedback about whether or not their trajectories were actually within specification[4]. In the **AL-WF** condition runtime verification (using code from Cardona et al. [8]) ensured that participant's provided trajectories matched the specification. Here, it is possible for users to get stuck if they are not able to think of trajectories that meet specification, since we do not let them proceed with specification-violating plans, unlike in **AF-NF**. Therefore, users were had the option to give up on the question after three consecutive failed attempts at trajectory generations, and are asked to provide a guess. Guessed answers were denoted as incorrect for the purpose of scoring.

A **control** group performed the same task as the test groups, but were not required to provide trajectories for their validation process nor were they given the option to do so on the interface. All users were provided with scratch paper, and it is possible that a subset of control users may have essentially executed a similar workflow tracing on-screen with their finger or on scratch paper.

Finally, participants were asked to comment on their approach to answering questions, their overall confidence, the presentation of specifications, and any features of the interface which helped or hindered their understanding. Participants could also provide free-response commentary on their thoughts on the experiment as a whole.

## 4.2 Experiment Hypotheses

Here, we introduce our hypotheses, metrics to assess each hypothesis, and statistical procedure.

Our hypotheses are:

- **H1:** Active learning increases the rate of correct responses in specification validation. This hypothesis is assessed by comparing the overall accuracy across conditions **AL-NF** and **AL-WF**, to the control.
- **H2:** Active learning leads to better calibration of confidence rating with response correctness in specification validation. This hypothesis is assessed by comparing the correlation coefficient between confidence and validation accuracy across conditions **AL-NF** and **AL-WF**, to the control.
- **H3:** Increased rate of adherence to specifications in trajectory generation is correlated with increased rate of correct responses in specification validation in the **AL-WF** condition. This hypotheses is addressed by comparing the per-question rate of satisfactory trajectory generation with response correctness in the **AL-WF** condition.

Omnibus tests were conducted to examine the relationship between various subject-specific and specification-specific predictors and subjects' success at the validation task. The statistical models and further analysis procedures are detailed in Appendix A.5. Additionally, we explicitly measured participant engagement and exclude users who "gave up" in our analysis. These representative cases were determined via criteria given in Appendix A.7 and indicated users did not actively interact with the experiment content. See Section 5.2 for more details on subjects' engagement.

# 5 Results

## 5.1 Validation Task Performance

In assessing **H1**, we find that there was no significant difference in the participants performance among these groups (Table 1, $F(2, 42) = 0.0804, p = 0.453$). Bartlett's test for homogeneity of variance in score across learning conditions revealed no significant differences ($p = 0.744$). Subjects' performance at the validation task was significantly better than random across all conditions (Table 1), showing some minimal capability to validate system behaviors.

---

[4]A subset of four subjects from this experimental group were asked to perform experiment while speaking out loud to explain their thought process (the "think aloud" group). For these subjects, audio and screen recordings were taken, and their results were not analyzed with the rest of the subjects.

Table 1: The overall accuracy (mean and standard deviation) for the Validation Task, along with statistical significance when compared against random chance (score of 50%) and associated Cohen's $d$ effect size. Overall accuracy was $65 \pm 15\%$. **AL-NF** is active learning with no feedback, **AL-WF** is active learning with feedback.

| Condition | Accuracy | $p$ | $d$ |
|---|---|---|---|
| AL-NF | $66 \pm 16\%$ | 0.007 | 0.90 |
| AL-WF | $67 \pm 16\%$ | 0.001 | 1.00 |
| Control | $62 \pm 13\%$ | 0.009 | 0.86 |

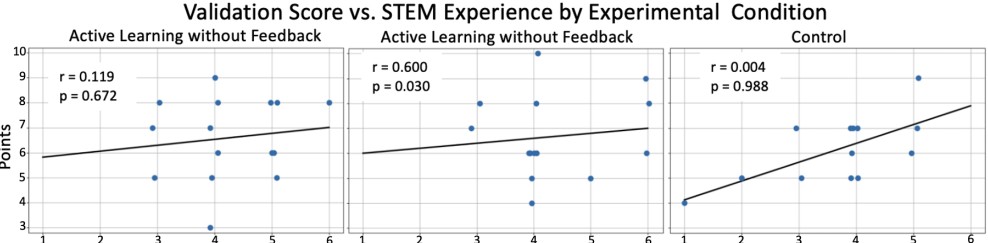

Figure 2: STEM experience vs. validation score by experiment condition. Horizontal jitter was added to visually separate points. Linear fit is shown. Displayed $r$ and $p$ values were calculated using Spearman's coefficient.

However, the data did not indicate significant differences in subjects' overall performance based on the experimental condition, participant level of education, STEM background, or formal methods familiarity (all $p > 0.05$). Further discussion of formal methods familiarity among our subject pool is provided in Appendix A.5. STEM experience appeared to be less influential in the active learning conditions than in the control (Figure 2). The correlation between STEM experience and performance was $r = 0.599$ and $p = 0.030$, though not significant due to the Bonferroni correction ($a = 0.05/3 = 0.016$).

The data did not support significant differences in question correctness based on factors of level of education, familiarity with formal methods, STEM Background, experiment condition nor question specific factors of specification ground-truth validity, specification complexity, nor the question sequence order as predictors of validation correctness (all $p > 0.05$).

In assessing **H2**, we find that the comparison of participants' average confidence levels between groups showed no significant difference ($F(2, 42) = 0.361$, $p = 0.699$). Further analysis of confidence and calibration between subjects' confidence and correctness can be found in Appendix A.6.

## 5.2 Effective Engagement

Across each of the engagement mechanism conditions, some participants did not engage with the validation task (Table 2). Behavior trends for non-engagement or "giving up" were identified post-hoc by analyzing the duration of users' engagement with questions for all groups, as well as the content of responses in active learning conditions. We identified and excluded eight participants who were determined to have given up on the task ("giving up" criteria in Section A.7).

Table 2: Participant count and engagement. **AL-NF** is active learning with no feedback, **AL-WF** is active learning with feedback. Time to completion (mean and standard deviation) does not include participants who gave up.

| Condition | Number of Subjects | Number Giving Up | Time (minutes) |
|---|---|---|---|
| AL-NF | 18 | 3 | $36.31 \pm 10.85$ |
| AL-WF | 16 | 1 | $40.37 \pm 10.68$ |
| Control | 17 | 4 | $19.92 \pm 9.07$ |

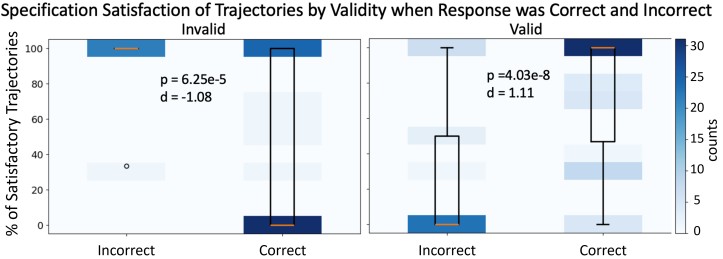

Figure 3: Combined heat map and boxplot (same data) showing satisfaction of specification in trajectory generation by correctness for valid and invalid ground truth. Each datapoint represents one response to a validation satisfaction question.

The two active learning conditions were not significantly different from each other ($p = 0.29$); however, they were both significantly greater than the control group ($p = 3.20 \times 10^{-5}$ and $p = 1.18 \times 10^{-6}$ for **AL-NF** and **AL-WF** conditions respectively, both less than the Bonferroni threshold). The active learning conditions effectively extended users time of engagement with the task; however, as observed in Table 1 this extended time was not effective in improving performance at the system validation task during the experiment session.

In **AL-NF**, certain users exhibited a tendency to fail the trajectory generation task by submitting trajectories that were completely uninformed by the specification (either of too short a length or not at all adherent to the constraints).

Under **AL-WF** participants were forced to engage for longer durations as they could not answer without providing at least one satisfactory trajectory. While 108 of 506 total provided trajectories were rejected for not satisfying the specification, only eight of the 160 questions were answered with a "give-up" response, and those responses came from only five of the 16 total participants.

## 5.3 Trajectory Analysis

Under **AL-NF** participants could provide trajectories which did not meet the specification. These trajectories were further analyzed to interrogate how users arrived at their answers. 34% of these trajectories did not actually meet the specification. Note that subjects were instructed to *only* provide trajectories that met specifications, and that it was always possible to do so. These trajectories violated the specification in a variety of ways ranging from simple off by one errors with the time bound, to complete ignoring of parts of the specification, to the creation of short, non-meaningful trajectories.

Participants' creation of trajectories that satisfy the specification is related to their correctness in the validation task, however this effect is dependent on the ground truth validity of the specification. When the ground-truth of the specification is *valid*, correct responses are more significantly more likely to be accompanied by trajectories which satisfy the specification (Figure 3, $p = 6.25 \times 10^{-5}$). Conversely, when the ground-truth of the specification is *invalid*, incorrect responses are more likely to be accompanied by satisfactory trajectories as shown work (Figure 3, $p = 4.03 \times 10^{-8}$). Proper trajectory generation indicates subjects' understanding of specification constraints, but the translation of this capacity to validation ability is not direct.

## 5.4 Subject Cognitive Process

Observations of subjects in the talk-aloud sub-group as well as the content of all subjects post-experiment comments provided insight in to the cognitive processes of the participants. In post experiment comments, 14 subjects (25% of the subject pool) expressed some degree of confusion and frustration with the process of system validation. Expressions varied in intensity from "it was a little complicated for me," to describing the process as "overwhelming," "disorienting," and "mind boggling." 57% of subjects who expressed such confusion participated in the **AL-NF** condition, five participated in the control condition and only one in the **AL-WF** condition. Seven participants expressed confusion about the presentation and syntax of specifications with four participants specifi-

cally noting trouble with nesting, despite the fact the specification complexity metrics (AST depth and symbol count) were not found to be a significant factor in determining performance.

Six participants explicitly expressed some concept of safety (avoiding getting tagged) and liveness (reaching the goal). A notable instance of this stated "I imagined my robot was "suicidal" and was trying to get tagged. If there was no way to be tagged, then it remained to see if it could reach goal in under 30 steps (the other way to lose)." The notion of the robot being "suicidal" suggests a consideration of safety, where the participant is cognizant of potential hazards that the robot may encounter during its operation, while the emphasis on reaching the goal within a specified time frame suggests an awareness of ensuring that the robot remains active and progresses towards its objectives.

# 6 Discussion

## 6.1 Validation Accuracy

These findings suggest that formal specifications are not inherently human interpretable for validation and that inciting further cognitive engagement with required trajectory generation is not a sufficient mechanism for improving interpretability over a single session. While all conditions proved better than random chance, the operational significance of accuracy rates of between 60 and 70 percent for system validation would be unacceptable for most operational use cases of automation.

Our results indicated while tasking users to perform trace generations meeting specifications as a step in performing system validation increases their time of engagement in the process, this step does not consistently improve task performance. While the validation question is asking users to consider all possible trajectories allowed by a specification the process of generation may overly draw their focus to a certain subset of the decision space.

The two active learning conditions may be improving the performance of participants with lower initial STEM experience (Figure 2) in comparison to the control, which would match previous work showing a dependence of active learning effects on initial levels of related experience [31, 30], but there are too few low-STEM-experience participants in our control group to test this hypothesis.

As noted by Vinter [39], individuals have difficulty working with formal specifications in myriad of ways and have a variety of preferences for specifications' verbosity. In our post-experiment commentary, 4 subjects noted trouble with operator nesting. The question with the lowest performance (32% correct) had a negation operator. Yet, other questions that included negation operators — with equivalent or lower length and AST depth — had typical correctness rates. While stringing concepts together, managing nested structures, and handling negation operators seem to present challenges, the impact of these factors on interpretability remains unclear. Our data do not provide sufficient support for conclusions about the causes or categories of these difficulties, though other works such as that by Booth et al. [3] examines the relationships between specific qualities of logical statements and human interpretability more directly. Exploration of various STL constructions on interpetability in conjunction with active learning practices could yield more nuanced insight into human comprehension of STL specifications.

Overall, the literature advocating for the use of formal methods for XAI seems to be dramatically underestimating usability issues, as well as the informational needs of users. We note, however, that our particular choice of system validation as the goal of interpretability places particularly high informational need on the human, as prediction of all consequences of the policies must be accounted for [36].

## 6.2 Giving Up

The forced engagement of active learning notably frustrated participants, matching with common student feedback from educational settings despite improved performance [34, 7], and in AI reliance studies where engagement elicited negative participant ratings despite more appropriate reliance levels [4]. The most expressions of frustration came from the **AL-NF** group, which may be because there is increased workload, but not increased feedback. In contrast, the minimal expression of frustration and confusion from the **AL-WF** condition may be due in part to the fact that frustrations were assuaged by the positive feedback subject received with successful saving of trajectories and

hints provided with negative feedback of failed saves. This result hints at a way to improve user engagement and minimize frustration while maintaining potential active learning benefits.

### 6.3 Limitations

A limitation of this work is the relatively abstract setting of the validation scenario. In order to construct a concrete test case, we used a grid world and took a set of "game-winning conditions" as the participants' intent, acting as though they could not be easily codified. Having a specific, simple set of "user intents" was required to ensure the objectivity of the validation ground truth. A real-world setting would likely have more complex world dynamics, but also a more ill-defined set of stakeholder intentions that are actually difficult to translate appropriately to policies.

Another limitation, and a difference between this work and much of the active learning literature, is that we are limited to a single session of learning and testing. Studies exist that both show [41] and do not show [33] a positive effect of active learning in a single session, and much of the work in this space showing strong effects is over the course of a semester or more [1, 16]. It may be that active learning only shows a difference with the control condition over those periods, so multi-session studies should be conducted in the future.

Finally, our participants had a range of experience with formal methods, and included many complete novices, which would presumably not be the case with any operational validation scenario. However, previous work showed that formal methods expertise increases confidence in validation substantially, but only increases accuracy slightly, and critically, decouples accuracy from confidence [38]. A more useful demographic to have here would likely be participants who are familiar with the task domain, though as previously noted, that would require a more defined validation task.

### 6.4 Implications and Recommendations

An observation about the approach towards validation is that the thought process is one of predicting and finding edge cases: situations in which the requirements are met, but the intent is violated. The more aligned the two are, the rarer these cases. This framing perhaps points to ways to ameliorate the situation, which some participants did indeed realize (Section 5.4). On the human side, priming and training in finding cases such cases may be useful, which may be explored in future work. Future experiments might also consider whether whether people trained in finding edge cases (e.g. lawyers, compliance officers, forensic accountants, etc) may perform better than laypeople or formal methods experts in checking formal policies. From the machine side, we may consider whether procedures that automatically show edge cases to human examiners may be helpful. Here, STL's quantitative semantics are attractive, as a diverse set of minimally-robust trajectories could represent conceptual edge cases that are worth examining, and may be generated automatically.

Similar to [38], we again find that the common claims of formal methods interpretability to be unfounded for the issue of validation. Indeed the continued claims of "interpretable" methods without definition or evidence is akin to claiming "accurate" learning methods without providing accuracy measures. These unsubtantiated and poorly-defined claims can have significant negative societal impact, as both researchers and potential users are driven towards unsupported methodologies. We recommend that the community define their claims more specifically, and provide evidence that the claims are met. At the same time, it is clear that interpretability cannot simply be considered from a technological perspective — the needs of the user in the specific task context are of paramount importance, and is an underexplored area where significant improvements can be made [36].

## 7 Conclusion

This study is the first to our knowledge that attempts to unite research in human learning to formal specification validation. Under conditions of just examining a specification, using active learning via trajectory generation, or using active learning with feedback on generated trajectories, all conditions are significantly better than random, but no condition is significantly different from any other, with an overall validation accuracy of $65\% \pm 15\%$. These results continue to call into question common claims that formal specifications are inherently human-interpretable. Further intervention and design of validation procedures on both human and machine sides are necessary to understand what methods actually work for the validation task, a requirement for robot reprogramming after deployment.

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

# A    Appendix

## A.1    Signal Temporal Logic for Programming Autonomous Systems and Representing Behavior

The Signal Temporal Logic (STL) language specifies temporal properties of real-valued signals using signal predicates. A signal predicate $\mu$ is in the form $f(x(t)) > a$, where x(t) represents a signal that must follow the conditional specified within the predicate (i.e., larger than $a$). The syntax of signal temporal logic (STL) is formally defined as an aggregation of these primitives and temporal operators:

$$\varphi ::= \mu|\neg\varphi|\varphi \wedge \varphi|G_I\varphi|F_I\varphi|\varphi U_I\varphi \tag{1}$$

Here, the range $I = [a, b]$ represents a time interval where the logic must hold True. $G$ represents the Global operator, $F$ represents the Finally operator, and $U$ represents the Until operator. Formally,

- At time $t$, if $G_I(\varphi)$ holds, then $\varphi$ holds $\forall t$ in $t + I$.
- At time $t$, if $F_I(\varphi)$ holds, then $\varphi$ holds at some $t'$ in $t \in t + I$.
- At time $t$, if $\varphi U_I \varphi'$ holds, then $\varphi$ holds at some time $t' \in t + I$ and $\forall t'' \in [t, t')\varphi'$ holds.

Given an STL specification (i.e., a composition of formula and operators), a set of acceptable robot behavior can be created. Our paper is concerned with helping users understand the set of possible behavior and allowing them to ensure whether this set meets their specifications. For simplicity, we only use the *F* and *G* operators in this study, and not the *U* operator.

## A.2 Specification Specifics

All specifications were expressed as location constraints. The basic structure of these constraints involved spatial variables X and Y, representing grid squares in the ManueverGame 2D gridworld. The relational operators $<, >, \leq, \geq,$ and $=$ were employed to define spatial relationships. These atomic propositions were then combined using the following logical and temporal operators propositional logic operators: AND (Conjunction), OR (Disjunction), NOT (Negation) and temporal operators ALWAYS and EVENTUALLY. Time constraints were defined using language, specifically stating the intervals with the phrase "from time A to time B" for each temporal predicate. The logical and temporal operators were expressed in their linguistic form rather than symbolic notation. The resulting specification was ultimately of the form of STL, but with some increased clarity to those unfamiliar with STL.

All specifications were formulated as absolute positions rather than relational ones, which was intended to make the task nontrivial (otherwise, statements such as "Eventually distance from goal = 0 AND Globally distance from hostile > 2" could be used). The complexity of specifications was varied and measures of AST depth and specification length were captured. However, we did not produce further classification of specification difficulty due to the ambiguity inherent to this this task. A complete list of the specifications presented to subjects is provided in the supplementary material.

## A.3 Subject Introduction Process

Subjects were first introduced to the experiment through a series of short video clips that described the experiment flow, accompanied by a supporting text-based tutorial. Subjects were introduced to the *ManeuverGame* interface and the rules of the game through a web-based tutorial integrated with the *ManeuverGame* software suite.

The introductory material gives clear visualizations of winning conditions and various loss conditions (the robot not reaching the goal within the time bound or the robot being "tagged" by the opponent).

The introduction section also provided explanations of STL notation and examples of creating trajectories that satisfy STL specifications. Concept check questions were provided to ensure participant understanding of the game mechanics, specification notation and the validation task. Additionally, practice validation questions were provided to give users practice interacting with the *ManueverGame* interface and experiment validation procedure.

## A.4 Formal Methods Familiarity Representation and Analysis

Responses to the question of formal methods experience were recoded in instances where participants' open-ended explanations showed that they did not actually understand what formal methods meant. In these instances we exclusively reduced the level of coded familiarity.

Formal methods familiarity was not found to be significant with either the original nor recoded values in the omnibus tests nor the post-hoc rank order correlation in Figure 4. However, formal-methods experts were underrepresented in our subject pool with the mean and standard deviation of participants' self-reported familiarity rating after recoding being 2.00 out of 5 $\pm 1.26$ and only 7 participants with self-rated familiarity > 3.

## A.5 Statistical Models and Analysis

A mixed-effects regression examined the participant based predictors of familiarity with formal methods, STEM experience, level of education and experiment condition with overall validation performance serving as the response variable. The regression was followed by independent two-sample t-tests for categorical variables, or a Spearman's correlation for continuous variables if the omnibus returned significance.

Omnibus logistic regression analysis was performed on a by-question basis with participant-based predictors of familiarity with formal methods, STEM experience, level of education, experiment condition as well as specification-based predictors of specification ground truth validity, question sequence number, specification AST depth, specification symbol count and a response variable of validation correctness.

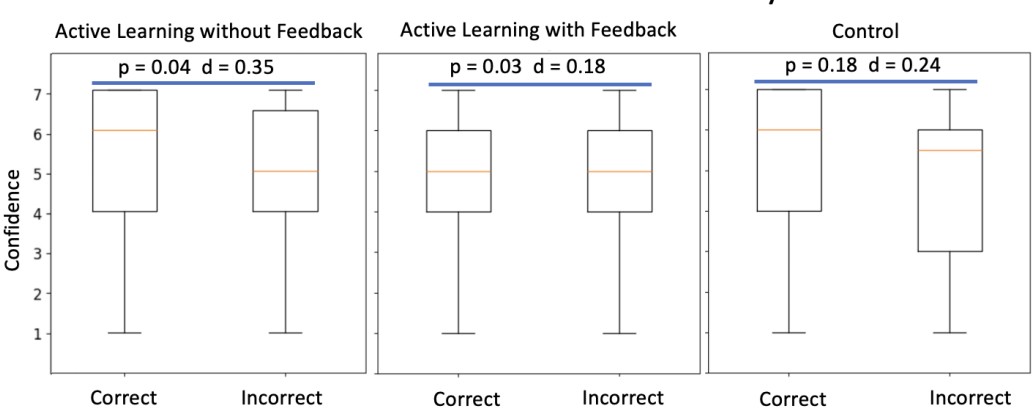

Figure 4: Formal Methods Familiarity vs. Validation Score. Horizontal jitter was added to visually separate points. Displayed $r$ and $p$ values were calculated using Spearman's coefficient but a linear fit is shown.

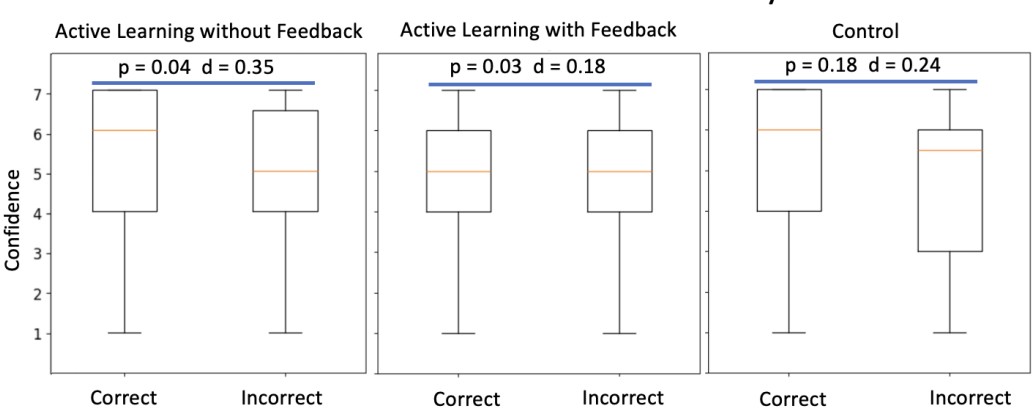

Figure 5: Participant confidence in their answer when their answer was actually correct vs incorrect, split by experimental condition.

### A.6 Confidence Analysis

An examination of variance equality for confidence ratings yielded a non-significant result ($p = 0.313$). Independent t-tests between confidence values when responses were correct or incorrect (Figure 5) indicated users' confidence was not significantly different based on the correctness of their answer in any of the conditions. The effect size of the active learning without feedback condition is moderate though not significant with the Bonferroni threshold (Figure 5, $p = 0.040 > 0.05/3 = 0.016$). Active learning without feedback could be a promising mechanism to improve users' confidence calibration.

Participants' average score across the entirety of the experiment also did not appear to be significantly correlated with their overall performance (7. Note the correlation between confidence and overall performance for control group was moderately positive though not significant with the Bonferroni correction (Figure 7 $r = 0.55$, $.05 > 0.05/3 = 0.016$).

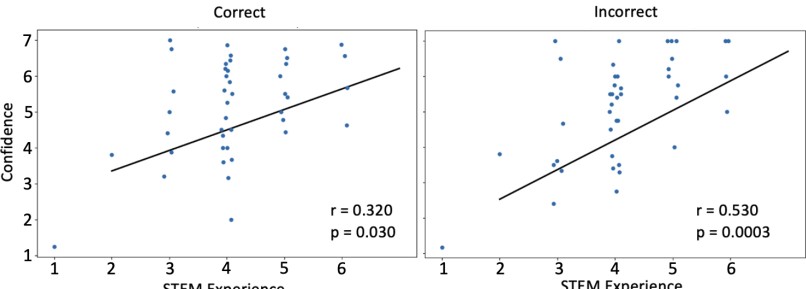

Figure 6: Participants' Average Confidence vs. STEM Experience when Correct and Incorrect. Horizontal jitter is shown to visually separate points. The correlation was calculated using Spearman's coefficient, but a linear fit is shown.

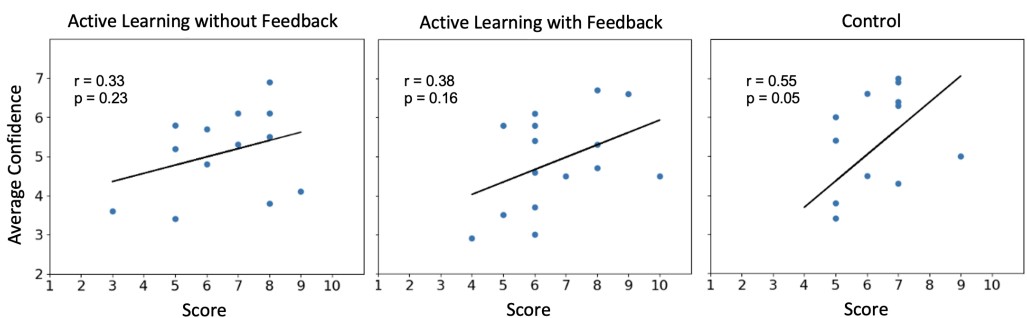

Figure 7: Subject's average confidence across all questions vs total validation score for each of the experimental learning conditions. A linear fit is shown but correlations were calculated using Spearman's coefficient. All of the correlations were positive but none were found to be significant with the Bonferroni correction ($a = 0.05/3 = 0.016$).

There is a significant positive correlation between STEM experience and average confidence both when the question was answered correctly and incorrectly (Figure 6). The correlation is stronger on questions that were answered incorrectly (Figure 6, $r = 0.530$, $p = 0.003$) compared to when answered correctly (Figure 6, $r = 0.320$, $p = 0.03$). Those with experience in STEM fields may be more confident in their responses, but such confidence is substantiated by better performance.

## A.7 Giving Up

The determination for giving up was based upon users time spent completing the experiment as well as their interaction with the experiment interface. Due to the differing requirements on subjects, different criteria were used to determine who gave up. The following criteria were applied to subjects in the three conditions:

**Control:**

- < 1 minute spent per question on average

**Active learning (no feedback):**

- < 2 minutes per validation question on average OR
- at least 3 questions were answered with extremely short, nonsensical trajectories (e.g. < 3 time steps)

**Active learning (with feedback):**

- < 2 minutes per validation question on average OR
- choosing to give up (on the interface) on three or more questions

