# OpenReview forum: "STL: Still Tricky Logic (for System Validation, Even When Showing Your Work)"
_NeurIPS.cc/2024/Conference — NeurIPS 2024 poster_

### Official Review · Reviewer_d2Sv · 2024-07-07

**Soundness:** 3
**Presentation:** 3
**Contribution:** 2
**Rating:** 5
**Confidence:** 2

**Summary:**

The paper investigates the efficacy of formal specifications (specifically, Signal Temporal Logic (STL)) for human validation of autonomous systems. The authors distinguish between *verification* (whether an implemented policy adheres to a formal specification of its behaviors) and *validation* (whether the system's specifications align with higher-level goals). They correctly assert that validation is an inherently human-centered and subjective process. While many existing explainable AI methods increase the system's explainability, humans still struggle to validate robot behaviors specified through formal methods. The main question addressed in the paper is *whether active learning methods can improve human performance in validating formal specifications.*

The paper conducts a comprehensive user study (n=55), dividing participants into three groups: no active learning (control), active learning with no live feedback (AL-NF), and active learning with live feedback (AL-WF).

The main conclusions can be summarized as follows:
- There is no significant improvement in validation accuracy with active learning.
- Active learning increases user engagement, and active learning with feedback reduces user frustration.
- Active learning appears to have a greater impact on participants with lower STEM experience (sample size was however too small for definitive conclusions.)

**Strengths:**

1. The paper correctly states that most human users are not capable of validating system behaviors using formal temporal specifications. It raises an interesting question about whether active learning methods can improve this capability. The experimental results are valuable as they show no significant improvement over the control group with active learning, which is surprising and definitely calls for further research at the intersection of human learning theories and formal system specification methodologies to bridge this gap.

**Weaknesses:**

1.  The main weakness of the paper is the mismatch between the claims made in the introduction and the focus of the experiments. The introduction argues that formal specifications are not suitable for explaining whether a system achieves higher-level, human-centered goals. However, the experiments measure the capacity of human users to infer the achievability of formally specified goals, specifically whether an agent can safely reach a goal in 30 steps. This task can be automatically checked using model checking techniques, such as providing the STL specification to a model checker to verify goal satisfaction and produce counter-examples if not satisfied. Although the authors mention this misalignment in the limitations section and correctly state that it is unavoidable to keep the success criteria objective and machine-checkable, the significant mismatch makes it difficult to draw conclusions regarding the efficacy of formal specifications in validating human-centered, higher-level goals. A short discussion of this mismatch in the introduction or later sections would have been beneficial.

2. The introduction and related works sections do not discuss the state-of-the-art techniques for enhancing the validation process of formal temporal specifications. There is already work on improving the interpretability of temporal logic specifications, such as translation to natural language [1] and hierarchical specifications [2].


[1] Cherukuri, H., Ferrari, A., Spoletini, P. (2022). Towards Explainable Formal Methods: From LTL to Natural Language with Neural Machine Translation. In: Gervasi, V., Vogelsang, A. (eds) Requirements Engineering: Foundation for Software Quality. REFSQ 2022. Lecture Notes in Computer Science, vol 13216. Springer, Cham. https://doi.org/10.1007/978-3-030-98464-9_7

[2] X. Luo and C. Liu, “Simultaneous task allocation and planning for multi-robots under hierarchical temporal logic specifications,” arXiv preprint arXiv:2401.04003, 2023.

**Questions:**

- Line198: Why the guessed answers were scored as incorrect? This seems to ignore potential role of feedbacks in increasing the chances of making a correct guess.

**Limitations:**

[As I mentioned in the Weaknesses section] The main limitation of the paper is that the experiments define system success as "whether an agent can reach a goal in 30 steps", which is a specific and objective task that can be verified through formal methods like model checking. In contrast, the introduction suggests an interest in higher-level, subjective human-centered goals, which are not directly addressed by the experiments.

---

> ### Author Rebuttal · Authors · 2024-08-06
>
> High Level Objectives and Experiment Setup (Weakness 1 and Limitation)
>
> We appreciate the reviewer's observations regarding the perceived incongruity between the high-level objectives delineated in our introduction and the specific experimental task employed in our study. Upon reflection, we recognize that there is an opportunity to further elucidate the distinction between verification and validation processes, and better explicate how our experimental paradigm aligns with real-world workflows and the broader implications of validation procedures.
>
> The experimental task, which involves determining whether a given specification will result exclusively in trajectories allowing the agent to safely reach a goal within 30 steps, can be conceptualized and solved as a verification problem. However, we deliberately framed it as a validation problem, wherein the "game rules" of reaching the goal within 30 steps and avoiding the opponent serve as proxies for higher-level, human objectives. This methodological decision was made to facilitate an objective assessment of our hypotheses.
>
> We acknowledge in the present paper that the utilization of these rules as surrogates for more complex, human-centered goals renders this experimental task equivalent to a verification task which could be solved with verification techniques such as model checking (lines 151-158). Nevertheless, our framing as a validation problem within the experimental setup allows us to explore the nuanced differences between verification and validation processes, particularly in contexts where system goals may be less explicitly defined or more subject to interpretation. This reframing enables us to investigate the cognitive processes and decision-making strategies employed by participants when confronted with a task that, while structurally similar to a verification problem, requires a more holistic evaluation of system behavior against broader objectives. As we note in line 158, the "validation" task we present in this study is simple enough that failure to perform well in this context bodes poorly for when it is to be replaced with a less explicitly-defined validation. This approach aligns with the validation challenges often encountered in real-world scenarios, where the assessment of system performance extends beyond mere compliance with specifications to encompass the fulfillment of overarching goals and stakeholder expectations. This work lays important groundwork for future research in bridging formal methods and human-centered validation processes.
>
> Theoretical Interpretability Improvements vs. Empirical Studies (Weakness 2)
>
> We appreciate the reviewer providing further background works. Our main focus in this work is to fill in the gap in interpretability studies for STL validation that use real human validators. While Cherukuri et al. provides a mechanism for translation into natural language and evaluation of the translation quality with BLEU, we do not find such techniques to be within our scope on account of the fact that their claims for interpretability are not being tested with human subjects thinking through logical implications, but rather with natural language translations, which are not equivalent. Moreover, language translations of formal logic have not been shown to be an effective mechanism for improving human interpretability. Vinter et al. (1996) and Vinter (1998) showed that specifications rendered in natural language (even when containing logical operators) evoked inappropriate systemic biases in which readers substituted heuristic reasoning (commonly used in language) for logical reasoning (necessary for formal methods) during evaluation. This again highlights the difference between simply translating a specification back and forth between natural and logical languages, and correctly understanding the implications of the specifications (in either form). Neither rendering of the specification guarantees appropriate understanding.
>
> The hierarchical structure for logic specifications presented in Luo and Liu offers an intriguing approach to formal method interpretability; however, the lack of empirical evaluation with human subjects limits our ability to assess its practical efficacy within the context of this work. Moreover, while this work effectively reduces the length of individual formulas, its applicability may be limited in contexts such as ours where formulas are already concise with at most total symbol count under 100 and at maximum 7 temporal clauses.
>
> We appreciate the reviewer pointing us to recent work on improving the interpretability of temporal logic specifications, but must note that unlike the references we present in lines 38-41, alongside the Vinter work, the reviewer's references involve no human evaluators --- a requirement for making strong claims about human-interpretability. The Cherukuri and Luo works may very well be helpful in improving interpretability, but human evidence of their efficacy is yet to be seen.
>
>
> Question on scoring guesses (Question 1)
>
> The opportunity to guess was only provided to subjects after they choose to "give up" and told that their response would be recorded as incorrect. This opportunity was provided following multiple failures to provide trajectories that met the specification as described in lines 196 and 197 of the manuscript.
>
> Limitation
> (See response to High Level Objectives and Experiment Setup.)

---

> > ### Comment · Reviewer_d2Sv · 2024-08-12
> > **Thanks for your response.**
> >
> > Thank you for your response. All of my questions have been answered, and I don't have any further inquiries. I am maintaining a score of 5. My main concerns remain the limited scope of contributions and the paper's suitability for NeurIPS.

---

> > > ### Author Response · Authors · 2024-08-12
> > > **Re: suitability for NeurIPS**
> > >
> > > Thank you for the comment. Since the suitability for NeurIPS is a new concern that was not brought up in the initial review, we would like to address that issue specifically. As we responded to reviewer 14rd when the same suitability concern was stated:
> > >
> > > "...our work tackles an important topic that is ignored by many studies in logic: validation of behavior via interpretable specifications. The implications of our work relate broadly to logic and XAI, and future work in this area will improve human-AI/human-robot interaction with logic-based systems. NeurIPS specifically has had a variety of work that employs formal logic, such as the following:
> > >
> > > Differentiable Learning of Logical Rules for Knowledge Base Reasoning by Yang et al. (NeurIPS 2017)
> > >
> > > Logical Neural Networks by Riegel et al. (NeurIPS 2020)
> > >
> > > Interpretable and Explainable Logical Policies via Neurally Guided Symbolic Abstraction by Delfosse et al. (NeurIPS 2023)
> > >
> > > As these works (among others) make interpretability claims about logic-based systems (and were accepted to NeurIPS), our work in empirically checks these claims, is essential. Quoting Professor Michael Carbin from the Charles Isbell NeurIPS 2020 keynote, "The issue is not just correctness, but understanding the problem across the entire pipeline to understand what correctness is," which strongly mirrors our quotation of Professor Nancy Leveson (lines 39-41)."

---

### Official Review · Reviewer_14rd · 2024-07-09

**Soundness:** 2
**Presentation:** 3
**Contribution:** 1
**Rating:** 4
**Confidence:** 3

**Summary:**

This paper studies the claim of Signal Temporal Logic (STL) specifications being human interpretable and provides results from an experiment with human participants studying a potential active learning technique to improve explainability metrics. Results show that while human engagement is improved, system validation score changes are negligible.

**Strengths:**

The paper is written well and tackles an important topic that is ignored by many papers in STL and other temporal logics which claim human interpretability.

**Weaknesses:**

- The reviewer feels that this study, while important, may be a better fit for a more specialized venue.
- The paper provides a study on the (mostly) unhelpful effects of active learning on the performance of the subjects (Table 1) but does not clearly indicate the usefulness of this knowledge or whether it significantly contributes to the community beyond what is already presented in [15, 33].

**Questions:**

1. While the claim that STL is directly human interpretable can be brought into question, does it provide a useful middle ground towards interpretability for task descriptions? Could an STL specification be fed into a Large Language Model (LLM) tool to yield a description more favored by human participants? To this reviewer, intuitively,  a temporal logic specification is often more readable than any general neural network policy (viz. model parameters) or say a few “desired” trajectory samples.
2. The example specification in Fig. 1 has several different time intervals considered (4 to 12, 15 to 30, etc). This may hinder “human interpretability” significantly as the user may need to parse the trajectories multiple times. Have different measures of difficulty been considered in the study such as a common set of time periods among the specifications or reduced number of AND clauses? A description of these difficulty classes may yield answers towards what parts of STL are difficult for the users.

**Limitations:**

- Primarily stated as a “negative result” paper, while showing that active learning may not help STL interpretability scores, a solution is not discussed.
- STL specification difficulty classes were not defined or considered.

---

> ### Author Rebuttal · Authors · 2024-08-06
>
> Fit for NeurIPS (Weakness 1)
>
> As the reviewer notes, our work tackles an important topic that is ignored by many studies in logic: validation of behavior via interpretable specifications. The implications of our work relate broadly to logic and XAI, and future work in this area will improve human-AI/human-robot interaction with logic-based systems. NeurIPS specifically has had a variety of work that employs formal logic, such as the following:
>
> - Differentiable Learning of Logical Rules for Knowledge Base Reasoning by Yang et al. (NeurIPS 2017)
>
> - Logical Neural Networks by Riegel et al. (NeurIPS 2020)
>
> - Interpretable and Explainable Logical Policies via Neurally Guided Symbolic Abstraction by Delfosse et al. (NeurIPS 2023)
>
> As these works (among others) make interpretability claims about logic-based systems (and were accepted to NeurIPS), our work in empirically checks these claims, is essential. Quoting Professor Michael Carbin from the Charles Isbell NeurIPS 2020 keynote, "The issue is not just correctness, but understanding the problem across the entire pipeline to understand what correctness is," which strongly mirrors our quotation of Professor Nancy Leveson (lines 39-41).
>
> Contributions Beyond Greenman et al and Siu et al (Weakness 2)
>
> Greenman et al. is a study performed in a school setting with people trained in formal methods, seeking to understand misconceptions in LTL. While formal methods experts are important stakeholders in the validation process, they are not the only ones who need to understand the implications of specifications. Further, that work is concerned with general misconceptions, rather than the case of system validation.
>
> Siu et al. involved subjects with varying levels of familiarity with STL, but focused on methods that the formal methods and AI community believed were interpretable --- raw formal logic, decision trees, and natural language. That contrasts with the present study as we focus on methods demonstrated by the educational community (lines 104-119), heeding Miller's argument that the XAI community ought to draw from the work of experts in human learning in building our methods. Like Siu et al, our negative result calls into question the frequent claims being made about formal methods interpretability without evidence, and highlights the need to providing empirical evidence to support these claims, but we do so from the perspective that Miller takes.
>
> Translation into Natural Language (Question 1)
>
> The idea that a language translation might improve human interpretability seems intuitive, as the reviewer notes. However, much like the "intuitive" interpretability of other XAI methods that were not tested with users, close examination of the user study literature showed results to the contrary. We did not explore language translation because earlier findings with user studies (Vinter et al., 1996,  Vinter 1998) and not simply intuition, showed that specifications rendered in natural language evoked inappropriate systemic biases where readers substituted heuristic reasoning (commonly used in language) for logical reasoning (necessary for formal methods).
>
> Complexity of Time Periods (Question 2)
>
> We agree that varying time intervals and multiple clauses may impact interpretability. We did not consider this kind of complexity, but note that in practice, creating any kind of autonomous system that only uses a single time interval or a minimal number of AND clauses severely limits the range of potential systems under consideration.
>
> Negative Result without a Demonstrated Solution (Limitation 1)
>
> Many papers in temporal logic assume human interpretability without empirical human evidence. While this work is a negative result, future work includes expanding to a multi-session setup in order to better resemble the structure of active learning pedagogy in classroom environments and perhaps also a focus on compliance based professions (lawyers, compliance officers etc.) who may have better training at identifying edge case failures. Moreover, future work should investigate how other machine-based solutions, such as automatically highlighting edge cases, could aid users in developing an accurate mental model of the limits of a specification. We refrain from endorsing a specific solution at this stage, emphasizing that thorough human subject studies are crucial to substantiate any claims about improving interpretability. Pushback against overreach of claims is a natural and appropriate part of the scientific discourse, as argued by in the XAI case by Miller et al, (2017) [28], particularly when others are not checking the claims as we do.
>
> STL Difficulty Classes (Limitation 2)
>
> We defined and considered specification complexity by the number of symbols (43 to 97) and the abstract syntax tree depth (3 to 5) (lines 182-184). The data did not support that these factors affected performance (line 242). We could have codified difficulty into classes, though how to classify these is unclear, and we are already using the field's standard measures of complexity. For context, the example in Figure 1 has 46 symbols and an AST depth of 3, an example that is on the simple end of our complexity spectrum. A full set of our specifications and maps is in the supplementary PDF.
>
> As noted by the Vinter studies, individuals have difficulty working with formal specifications in different ways and have a variety of preferences for specifications' verbosity. In our post-experiment commentary, 4 subjects noted trouble with operator nesting. The question with the lowest performance (32% correct) had a negation operator. Yet, other questions that included negation operators --- with equivalent or lower length and AST depth --- had typical correctness rates. This underscores the nuanced differences in how participants engage with formal specifications and the complexities associated with various specification constructions.

---

> > ### Comment · Reviewer_14rd · 2024-08-11
> >
> > I thank the authors for their response and their rebuttal PDF explaining the specifications considered. Based on reading the other reviews as well, I am optimistic that the community will be interested in the results presented.
> >
> > However, I am still uncertain whether the presented study, in its current form, will be useful to XAI researchers without further in-depth exploration of what makes certain parts of STL challenging for human evaluators. Since the presented work is a step in the right direction, I will raise my score accordingly. Nevertheless, I remain skeptical about how useful the work is compared to Siu et al. [33], given that STL was a subset of the logic descriptions they considered.

---

> > > ### Comment · Area_Chair_72dW · 2024-08-12
> > > **Re: Comment on Siu et al.**
> > >
> > > Dear Authors,
> > >
> > > Could you comment on how you would differentiate your work from [33] Siu et al.? This would really help future discussions.

---

> > > > ### Author Response · Authors · 2024-08-12
> > > > **Re: Comparison to Siu et al. 2023**
> > > >
> > > > We thank the reviewer and the area chair for the comments, and wish to clarify our contributions in comparison to Siu et al. 2023.
> > > >
> > > > Siu et al 2023 and this work are both concerned with the validation of STL formulas and both return negative results on humans' ability to validate such formulas to a high degree of accuracy. However, while Siu et al explored methods asserted as interpretable by the AI community - raw STL, decision trees, and natural language - this work takes a different approach with a far more involved procedure required of our human participants. One of Miller et al's [28] critiques of the XAI space was that it frequently proposed methods without grounding them in studies about how humans actually learn. That critique also applies to the methods used by the Siu et al. experiment, since those were based on the largely unfounded assumptions of the XAI community.
> > > >
> > > > In contrast, our active (human) learning approach (background in lines 105-119) is one that is based on the education literature, not the AI literature, and has been studied in human learning contexts for decades [1, 2, 13, 14], with [14] in particular showing that active learning increased learner performance and decreased failure rates across a metaanalysis of 255 studies on active learning (line 113). Active learning requires far more participant engagement than non-active learning. While what we present to participants is similar to Siu et al's experiment, Siu et al's experiment did not require participants to actually execute a related task and show examples of their thought process (lines 107-108). In contrast, our two active learning conditions (lines 185-198) required participants to demonstrate their understanding of the specifications by providing example trajectories that justified their validation decision, a task that was not part of the Siu et al. work. Essentially, while Siu et al's participants could just answer the validation question without thinking about it if they so chose, our active learning participants were forced to use methods that are shown by the education literature to improve human understanding.
> > > >
> > > > Particularly in response to the reviewer's concern about the contribution of this work vs Siu et al. given the similar findings: The fact that subjects' validation accuracy did not change sufficiently to be detectable for the task of system validation in this case likely speaks more to the difficulty of the interpretation and validation of specifications than it does to the efficacy of active learning (which again, has decades of empirical research behind it).
> > > >
> > > > There is significant differences between the present work and Siu et al.'s experiment, with the application of empirically-demonstrated human learning procedures being the fundamental driver. Siu et al's work was a step in the right direction of asking for empirical evidence of interpretability for validation, but by using methods established empirically in the human learning literature and not asserted in the AI literature, the present work shows to a much greater degree the depth of work required to make the claims of AI interpretability meaningful for this kind of use case.

---

### Official Review · Reviewer_67SW · 2024-07-11

**Soundness:** 3
**Presentation:** 3
**Contribution:** 2
**Rating:** 6
**Confidence:** 4

**Summary:**

The paper presents the results of a human study exploring the intuitiveness and interpretability of formal logic—in this case, signal temporal logic (STL)—in expressing policies for autonomous systems. Specifically, the authors study the effect of active learning, a pedagogical approach for human learning, on understanding the semantics of STL. The experimental results show that active learning, with or without feedback, provides no significant improvement, and STL remains a challenging logic for humans to understand.

**Strengths:**

+ The problem of understanding the role of formal logic in interpretable control is well motivated.
+ The experimental setup, ManeuverGame, and the corresponding logic STL seem appropriate for the study.
+ The user study appears to be well designed, with appropriate IRB approval.
+ The results are surprising, especially regarding moderate changes in the results with respect to the formal methods/STEM background of the participants.

**Weaknesses:**

- In the absence of a list of exact formal specifications used in the study, it is difficult to understand why participants found it challenging to validate the specifications.
- It is unclear whether the exercise tested participants' understanding of the domain or the logic.
- The exact challenges in understanding STL are not clearly discussed.

**Questions:**

- Would similar results be observed if natural language were used to express specifications? For instance, the specifications in Figure 1 do not have any STL-specific features that would complicate understanding.
- Is it possible that participants' performance resulted from a lack of familiarity with the experimental setup rather than with the logic?
- Which aspects of the logic do participants find challenging: modalities, time-tracking, or nesting of the operators?
- How did the authors choose the class of specifications? Are they known to be particularly challenging, or do they express some practically relevant requirement?

**Limitations:**

The paper presents a human study to better understand the role of STL in interpreting robot policies. The study involved three groups of participants, differing in the active learning received. The results show no major differences in performance among these groups. While the user study is clearly of interest, it is not clear if natural language would have been a better candidate for the same task. It appears that the key challenge lies in understanding a conjunction of requirements and their effects.

---

> ### Author Rebuttal · Authors · 2024-08-06
>
> List of Specifications Used (Weakness 1)
>
> We have included a full list of specifications and corresponding maps in the rebuttal's attached PDF. To clarify the nature of these specifications in the context of STL, we offer the following:
>
> All specifications were expressed as location constraints. The basic structure of these constraints involved spatial variables X and Y, representing grid squares in 2D gridworld. The relational operators $<, >, \leq, \geq,$ and $=$ were employed to define spatial relationships. These atomic propositions were then combined using the following logical and temporal operators propositional logic operators: AND (Conjunction), OR (Disjunction), NOT (Negation) and temporal operators ALWAYS and EVENTUALLY.
>
> All specifications were formulated as absolute positions rather than relational ones, which was intended to make the task nontrivial (otherwise, statements such as "Eventually distance from goal = 0 AND Globally distance from hostile > 2" could be used). While these specifications were intended to be relatively simple, the low rates of validation observed suggest that more intricate real-world systems, would likely encounter even more significant challenges. In examining the specific difficulties faced by subjects, a variety of failure causes were identified. Negation was highlighted as a significant challenge during the think-aloud portion of the experiment. Nesting was highlighted by 4 users in their post-experiment commentary as a challenge. Among the four specifications with the lowest performance (each with accuracy <50 percent), the main challenges presented by these specifications included time constraint alignment, operator coordination, negation handling, and nesting complexity.
>
> Subjects' Domain Knowledge vs STL Knowledge (Weakness 2 and Question 2)
>
> We acknowledge the concern about the experimental setup testing participant's understanding of the experiment domain or the STL logic. The experiment utilized a gridworld representing a capture-the-flag scenario, chosen for its familiarity to a wide range of users. To address potential gaps in understanding, we included a comprehensive introductory section that thoroughly explained the game domain and its dynamics. This section typically required at least 20 minutes for most users to complete and featured a combination of text and animated explanations. This introduction not only provided text and animation explanations, but also interactive quizzes to ensure users correctly understood the game rules before they began gameplay. These quizzes evaluated users' understanding of movement through the gridword, all win and loss conditions, use of STL logic within the capture the game scenario, as well as use of the interface. All questions had to be correctly answered before subjects were able to progress through to the experiment task. Static versions of introductory material and quizzes are provided in the original supplementary material.
>
> Challenges in Subjects' Understanding (Weakness 3 and Question 3)
>
> With regards to exact challenges in subjects' understanding of STL, our experimental format, which evaluated participants on only 10 specifications in the course of a single session, did not allow for in-depth analysis of the specific challenges they encountered in understanding the specifications. The work of Vinter et al. (1996) and Greenman et al. [15] which address the cognitive challenges people encounter in working with formal logic included much longer periods of evaluation such as over the course of multiple academic years. We appreciate the reviewer's observation that "the key challenge lies in understanding a conjunction of requirements and their effects." It appears that many subjects struggled with evaluating the STL formulas due to misinterpretations of how different clauses, involving time and operator types, interact with one another. Future work expanding to a multi-session setup would allow for better understanding of specific challenges.
>
> Translation into Natural Language (Question 1)
>
> We acknowledge the reviewer's perspective regarding the potential benefits of translating formal methods into natural language for enhanced human interpretability. This viewpoint resonates with the initial intuition shared by the authors and many others in the field. However, we opted not to pursue language translation based on the findings from Vinter et al. (1996) and Vinter (1998). Their research revealed that specifications articulated in natural language—even those incorporating logical operators—can provoke systemic biases. Readers often revert to heuristic reasoning, which is common in everyday language use, rather than engaging in the logical reasoning that formal methods require during evaluations.
>
> Consequently, we believe that if natural language were employed to express specifications, subjects would likely make errors with similar or even greater frequency, albeit for slightly different reasons. The reliance on heuristic reasoning could lead to misinterpretations, undermining the very clarity and precision that natural language is intended to provide.
>
> Familiarity with Domain (Question 2) (See Subjects' Domain Knowledge vs STL Knowledge, above)
>
> Challenging Aspects of Logic (Question 3) (See Challenges in Subjects' Understanding, above)
>
> Choice of the Class of Specifications (Question 4)
>
> We chose the class of specifications and the scenario because gridworld movement (in our small environment) was something that we felt subjects could easily grasp. This assumption was validated by our pilot studies as well as by subjects' ability to answer questions in the introductory period that checked for their domain understanding. The simplicity of the validation task was a point of comparison against more complex and ambiguous validation tasks, which would be more difficult to perform. Poor performance here would likely indicate an inability to validate more real-world tasks (lines 155-158).

---

> > ### Comment · Reviewer_67SW · 2024-08-13
> >
> > I thank the authors for their responses to my questions. I am satisfied with the answers provided.

---

### Official Review · Reviewer_QXA3 · 2024-07-13

**Soundness:** 3
**Presentation:** 3
**Contribution:** 3
**Rating:** 5
**Confidence:** 3

**Summary:**

This paper examines the challenges of using Signal Temporal Logic (STL) for validating autonomous systems and finds that human validation accuracy remains loweven with active learning techniques. Using the ManeuverGame interface, the study tests three conditions—no active learning, active learning, and active learning with feedback—finding no significant improvement in validation performance. The research highlights the need for better validation techniques and human-computer interaction to enhance human interpretability and validation of STL-specified policies.

**Strengths:**

1. The study employs a well-structured experimental design using the ManeuverGame interface, which effectively simulates real-world validation scenarios. This allows for a thorough assessment of human validation performance across different conditions

2. The research provides valuable empirical evidence showing that active learning does not significantly enhance human validation accuracy, remaining around 65\%. This insight is crucial for understanding the limitations of current formal methods in practical applications.

3. The study highlights the cognitive challenges faced by humans in interpreting and validating STL-specified policies, pointing out areas for improvement in human-computer interaction.

**Weaknesses:**

1. In my understanding of formal method, if you want to verify a rather high-level specification (which I assume is what the authors mean by 'validate'): You either directly use a formal method enhanced synthesis for your policy (which is easily doable for their maneuver case study) or use model checking (which depends on what policy we care about), and no human supervision should be necessary. Could the author justify that?

2. I think the criticism against interpretability is unfounded: They are not showing how they are interpreting the formulas for the users (which they can certainly do better with existing online monitors) and I think this is unfair for people who actually spend a lot of time developing monitors that are interpretable (such as the robust online monitors). Plus, few papers on monitoring are cited, which makes the paper seem to lack of comparision.

**Questions:**

Please refer to the weakness part:

1. Justify why we need human supervision.

2. Explain the criticism against interpretability.

**Limitations:**

This is about user-study which has no potential negative societal impact.

---

> ### Author Rebuttal · Authors · 2024-08-06
>
> Human Supervision (Weakness 1, Question 1)
>
> We appreciate the reviewer's comment regarding the lack of necessity for human supervision in our experiment and acknowledge that model checking could be used given the the concreteness of our objectives. As described in lines 155-158 of the manuscript, we wished to simulate real-world scenarios where human judgment is indispensable, particularly with stakeholder intents that are not easily codified into formal specifications. In the text, we explicitly acknowledge the reviewer's concern, but note that the task is a stand-in for the aforementioned human judgement scenarios. Unfortunately, without presenting to the user a task that can be checked automatically, we cannot objectively check the user's ability to evaluate the specification. Participants' failure to correctly understand the specifications in the context of the task is indicative of the difficulty they would encounter in more ambiguous and/or complex cases (line 158).
>
> While methods like model checking and  synthesis are indeed powerful, they fall short in capturing complex, context-dependent intents for autonomous systems, such as "navigate safely" or "maintain user trust". Where verification asks the question  "does this product/behavior match with the set of requirements set out for it"... validation probes "does this product operate in and only in the ways that I want it to?" In our experiment, "winning" serves as an objective stakeholder intent that can be codified in programming. This stands in for the more complex and less easily codified intents that robotics stakeholders often have and allows us to explore the limitations and challenges of human validation in a controlled environment, providing valuable insights that are applicable to more complex, real-world tasks.
>
> For example, in an autonomous driving scenario it can be verified that a car is able to obey posted speed limits and this behavior can be verified in real time. However, validating that a vehicle is operating safely, obeying the flow of traffic, and properly responding to other drivers' intentions to merge are behaviors critical to its safe deployment that cannot so easily or directly be quantitatively checked.
>
>
> Critique of Interpretability (Weakness 2, Question 2)
>
> We acknowledge the point regarding the our critique of interpretability claims and recognize the important work of researchers developing robust online monitors, such as Deshmukh et al. (2017), Zhang et al. (2023). However, upon review of such work, we still do not find empirical evidence that users performing interpretation are able to do so well using these systems. Such evidence cannot simply be provided in a software-only context without human studies. If the reviewer can point us to human studies with users interpreting formal specifications that demonstrate system usability, we would be happy to incorporate them into our discussion.
>
> Our findings indicate a gap in testing these systems with human participants to support interpretability claims. While many works have these claims (e.g. line 89), few support them with evidence from humans studies. This is not to say that existing efforts do not present interpretable techniques, but rather that there is a pressing need to validate these claims of interpretability with human operators, as argued by Miller et al. (2017) [28].
>
> Moreover, our study included a monitor in the third condition (active learning with feedback) where the robustness of users' trajectories was checked when users marked them as complete. Trajectories with negative robustness could not be saved and users were alerted that their submission was invalid and prompted to retry. However, our subject pool did not show significant improvement in validation performance even with this support.
>
> We wish to emphasize the distinction between asserting a system's interpretability based on theoretical design principles and demonstrating it through empirical testing with human users. While theoretical claims rely on design principles to suggest interpretability, it is the empirical evaluation that provides concrete evidence of whether users can effectively understand and use the system. Even systems that appear to be interpretable at first glance, such as decision trees or translation into natural language, may not be actually interpretable, as shown in [33] as well as in Vinter et al. (1996) and Loomes and Vinter (1997), particularly for the difficult task of system validation, which requires understanding all potential edge cases.
>
> The evidence we found and cited (lines 38-41, 92-98), along with Vinter et al. (1996) and Loomes and Vinter (1997) which actually involved users, point to a preponderance of evidence that methods claimed to be interpretable in the academic literature are often not so when users interact with them. If the reviewer is able to point us to user studies that demonstrate the interpretability of monitors (or other formal methods tools) that are designed to be human-interpretable, we would happily consider them in the context of this experiment.

---

> > ### Comment · Reviewer_QXA3 · 2024-08-11
> >
> > Thank the author for answering my questions, my concern has been solved and. I have updated my score accordingly.

---

### Author Rebuttal · Authors · 2024-08-06

We thank the reviewers for their critical assessment of our work. As a note, we appreciate one of the reviewers pointing out that we did not include the example formulas and maps show to subjects. We have included this content in the rebuttal's attached PDF and will also include it in the final manuscript.

Contribution to XAI

This work builds upon Miller et al's (2017) work which advocates for the critical need to ground explainability work in cognitive science and human information processing [28]. Our study distinguishes itself not only within the formal methods community but also in the broader XAI field by leveraging well-established educational practices and theories. This approach offers a novel perspective on explainability, bridging the gap between AI systems and human comprehension.

Many interpretability studies claim human interpretability without providing empirical evidence, our work evaluates claims of human understanding with real humans. Despite producing a negative result, our study underscores the importance of empirical research involving human subjects. This outcome reinforces the understanding that intuitive solutions in XAI may not always align with actual human comprehension.


Why not natural language?

Although natural language is often perceived as more intuitive for human understanding, it also presents its own set of challenges and biases. For this, we point to the work of Vinter et al. (1996), Loomes and Vinter (1997), and Vinter (1998) which evaluated the use of natural language translations of formal logic. We will address this oversight and incorporate a discussion of their findings into the final manuscript. Ironically, the time in which Vinter performed his experiments was one in which natural language was the standard approach to specifications, and the general push in computer science was to move towards the use of formal specifications for greater human understanding.

We refer to the following works in our responses:

Loomes, Martin, and Rick Vinter. "Formal methods: No cure for faulty reasoning." Safer Systems: Proceedings of the Fifth Safety-critical Systems Symposium, Brighton 1997. London: Springer London, 1997.

RJ Vinter, MJ Loomes, and D Kornbrot. Seven lesser known myths of formal methods:
uncovering the psychology of formal specification. 1996.

Vinter, Ricky Jay. "Evaluating formal specifications: a cognitive approach." (1998).

Jyotirmoy V. Deshmukh, Alexandre Donzé, Shromona Ghosh, Xiaoqing Jin, Garvit Juniwal, and Sanjit A. Seshia. Robust online monitoring of signal temporal logic. Form. Methods Syst. Des., 51(1):5–30, aug 2017

Zhenya Zhang, Jie An, Paolo Arcaini, and Ichiro Hasuo. Online Causation Monitoring of Signal Temporal Logic. In Constantin Enea and Akash Lal, editors, Computer Aided Verification, pages 62–84, Cham, 2023. Springer Nature Switzerland.

---

### Decision · Program_Chairs · 2024-09-25

**Decision:**

Accept (poster)

**Comment:**

The paper investigates the question of the interpretability of STL formulas and identifies that they remain hard for users to correctly interpret. They also show that popular pedagogical methods like active learning don’t seem to completely correct the issue. The paper also shows how factors like STEM education do not seem to affect the outcome as strongly as one would expect. These results are interesting and relevant given the popularity of learned logical specifications to summarize agent behavior.

There were some common concerns that were raised by multiple reviewers. First off, multiple reviewers were thrown off by the obvious connection between the problem being studied and formal verification. After all, one could employ model-checking to verify if the system would ever invalidate the given specification. I would encourage the authors to provide a clear discussion of the goals of this study and how they differ from model checking. Then, there is the question of the influence of the form in which the formulas are presented. While natural language is one method, other forms could also be employed. The authors do point to some initial negative results, particularly in relation to natural language. It would be useful to consider this dimension as part of future work. Finally, there is the question of the significance of the results. While I do agree that a negative result on its own is quite useful, it might be worth considering why these issues arise in the first place and what a potential solution might look like. Or are the authors of the opinion that formal specification, at least STL, is a dead end as a method in XAI? A discussion along these lines might be a helpful addition to the paper.  This also connects to similarities and differences between the results in the paper and the one presented by Siu et al. 2023. The authors did provide a pretty detailed argument in their responses. I would recommend the authors also include a version of it in the paper.

In conclusion, I would recommend that the paper be accepted.